# Tourism and the Global Vectoring of Antimicrobial-Resistant Disease: What Countries Are Most Impacted?

**DOI:** 10.3390/antibiotics14111055

**Published:** 2025-10-22

**Authors:** Peter Collignon, John J. Beggs

**Affiliations:** 1Canberra Hospital, Garran, ACT 2605, Australia; 2Medical School, Australian National University, Canberra, ACT 0200, Australia; 3Independent Researcher, Melbourne, VIC 3000, Australia; beggsjo_home@yahoo.com

**Keywords:** antimicrobial resistance, water, food, One Health, travel, contagion, AMR, disease burden

## Abstract

**Background:** Tourists returning home and visitors from abroad often carry antimicrobial-resistant (AMR) bacteria. Many of these resistant bacteria are acquired from, or were spread via, the environment (especially water). Understanding the impact from acquiring resistant bacteria via tourism upon global antimicrobial resistance is limited. **Methods:** Traveller transmission of AMR bacteria can be estimated from combining the numbers of travellers with AMR bacteria rates in different regions and the prevalence of communicable diseases. We used resistance data (WHO and contemporary publications) to measure the prevalence of *E.coli* resistance to third-generation cephalosporins. The study uses data from 2019, the year with the most complete dataset that also predates disruptions to travel caused by the COVID-19 pandemic. We then used the global burden of disease study and travel data from the World Travel and Tourism to create regional and country level indices measuring the impact of AMR bacteria for 241 countries. Estimates of global travel patterns were obtained using a gravity-style trip distribution model. **Findings:** Regions with the highest impact of AMR bacteria from returning travellers were Northern Europe and Western Europe. The region with the highest impact of AMR bacteria from visiting travellers was the Caribbean where small island countries receive large numbers of visitors. For countries/administrative regions with populations greater than 5 million, the AMR bacterial travel impacts measured in decreasing risk order from the highest were Hong Kong, Denmark, New Zealand, Hungary, Norway and Sweden. **Interpretation:** For some countries the incidence of AMR infection among both visitors and returning travellers is much higher than in the domestic population. This impact and how these bacteria are acquired from the environment, must be factored into public health policies for containing global spread of AMR bacteria and as part of a One Health approach.

## 1. Background

Resistance and prevalence of disease are highly correlated with stages of economic development and the quantity and quality of social and health infrastructure [1,2,3]. Resistant bacteria in the environment, especially in water, can readily spread to many people and food animals [1,2,3,4,5,6,7,8]. People and animals that acquire resistant bacteria via water can then spread these bacteria directly to other people and/or animals or indirectly via foods and faecal material. Resistant bacteria can be acquired by drinking water in many countries, including municipal supplies [4,5].

Disease and resistant bacteria are problems that are not merely ones of domestic economic and health policy. Globalization and the unprecedented surge in the movement of people through travel, migration, healthcare transfers and conflict have become major vectors for the global dissemination of both established and emerging drug-resistant bacteria. These pathogens transcend international borders. Travelers to regions where antimicrobial resistance disease is prevalent risk becoming colonized and unwittingly transmitting these bacteria upon returning home. Similarly, visitors from such high-prevalence regions can introduce resistant strains into host populations [9,10,11,12,13,14,15,16].

An essential component of a One Health strategy to combat the global spread of AMR bacteria is identifying activities, geographic areas and sources (e.g., water) most responsible for generating and disseminating resistant bacteria, with human movement being a major factor [6,7,8]. This insight is vital for prioritizing resource allocation and informing targeted interventions. Countries with underdeveloped health, water and sanitation infrastructure are more likely to serve as reservoirs of resistant organisms, especially where water and sanitation systems are inadequate [1,2,3]. Rising global incomes and the declining cost of air travel have further amplified the role of travel as a conduit for AMR bacteria transmission [17].

Building a robust knowledge base is critical for assessing the impact and risks associated with travel between regions. Such knowledge not only informs better public health policy and guidance but also enhances awareness of AMR bacteria exposure risk, improving the timely diagnosis and management of related infections during or after travel.

Travel is a major part of the global economy and has important social and economic benefits. In 2023, the Travel and Tourism sector contributed 9.1% to the global GDP [17]. However, it also has the unintended consequence of helping to spread AMR disease. This paper matches global travel patterns to the resistant rates and infection rates in origin and destination pairs to estimate the size and direction of the disease vectors. To measure how human travel influences the spread of antimicrobial-resistant germs globally, data were assembled for 241 countries and semiautonomous territories, and a matrix of travel flows among all 28,920 possible country pairs was developed using a trip distribution gravity model.

## 2. Results

The likelihood of transmitting antimicrobial-resistant (AMR) bacteria between individuals is strongly influenced by two key factors: the underlying burden of communicable diseases and the resistance rates of those pathogens within each region. Regions with a high prevalence of infectious diseases provide more opportunities for AMR pathogens to emerge, spread and persist—particularly where healthcare infrastructure, safe water, sanitation and surveillance are limited. Figure 1 presents the global distribution of AMR rates used in this study, while Figure 2 illustrates substantial disparities in communicable disease prevalence across countries. Notably, low-income regions exhibit both the highest infectious disease burdens and some of the most limited capacities to mitigate AMR threats. These conditions not only elevate the local risk of transmission but also amplify the potential for international dissemination through travel and migration.

The relative impact of visitors versus returning travellers depends heavily on their numbers relative to each country’s population. Tourism is often concentrated in countries with significant historical, cultural, geographic or architectural attractions. This concentration distorts averages calculated across countries. On average, countries receive 39.2 visitors per 100 residents annually, while only 8.4 residents per 100 travel abroad. As a result, the overall pattern of exposure impact attributable to visitors tends to exceed that from returning travellers. Figure 3 illustrates the exposure impact posed by both groups, highlighting the importance of considering each in AMR bacteria-related policy discussions.

Although exposure impacts are calculated at the country level to capture granular differences, many of the 241 entities in our dataset include small island states, principalities and territories where data coverage is limited and population sizes are disproportionately small. In many cases, these countries resemble concentrated tourism hubs more than independent contributors to global antimicrobial resistance risk. For popular tourist destinations with small resident populations, the number of visitors can vastly exceed the number of residents, producing extreme visitor-to-population ratios that distort exposure metrics.

Despite introducing statistical outliers, these cases offer useful insight into how intense tourism pressures can elevate local AMR bacteria transmission risks. However, to enhance cross-country comparability and minimize distortion from data irregularities, our initial analytical focus is on regional patterns. Aggregating to the regional level smooths anomalies while preserving meaningful geographic variation and enabling clearer identification of global hotspots. These regional averages, shown in Table 1, provide a more stable foundation for interpreting the broader implications of international travel on AMR bacteria exposure risk. A value of zero means no risk from returning travellers or from visitors.

To complement the regional analysis, we next examine country-specific results to explore how national characteristics shape exposure risks more narrowly. The country-level global map of the AMR infection burden of travel is shown in Figure 4.

Visually, the country level data in Figure 4 exhibits strong regional clustering with the greatest impacts of travel being in the countries of northern and western Europe. Many factors will influence differences among countries within regions. For example, Australia and New Zealand exhibit similar travel patterns for inbound originations and outbound destinations, but the tourism sector is relatively larger in New Zealand which puts a greater AMR bacteria travel burden on New Zealand.

Focusing initially on countries with populations exceeding 10 million helps reduce noise from statistical outliers and provides a more stable basis for comparison. Table 2 presents AMR infection travel risks for this group, ranked in descending order. Notably, Sweden, France, Czech Republic, the Netherlands and Greece exhibit the highest exposure risks (ranging from 0.74 to 0.45).

## 3. Discussion

Area with poor infrastructure, especially with poor water and sanitation, have much higher rates of rates of resistant bacteria present in both people and in water [1,2,3,4,5,6,7,8]. The role of the environment as both a reservoir and means of transmission for then high-risk bacterial colonization of people in the host country is usually greatly underappreciated.

International travel subsequently plays a pivotal role in the global spread of antimicrobial resistance (AMR). Visitors from high-AMR regions can introduce resistant bacteria into host populations, while returning residents may import novel strains acquired abroad. That these risks exist is well documented [5,9,10,11,12,13,14,15,16], with the vast majority of travellers not consuming antibiotics or having contact with healthcare facilities. Thus, the resistant bacteria they then carry are most likely acquired from the “environment”. This study is the first to quantify these bi-directional risks at global scale, using comprehensive data on travel patterns and country-level antimicrobial resistance prevalence.

The opposite possibility can also occur with some visitors. When visitors from low AMR bacteria prevalence countries visit countries with a high AMR bacteria prevalence, then susceptible bacteria passed onto to others or into the environment should have, in the absence of antibiotic exposure, a higher fitness and could replace resistant microorganisms.

In Europe—particularly Scandinavia—returning travellers face one of the highest risks of AMR bacterial acquisition relative to their stay-at-home citizens. This phenomenon is notable given the region’s strong public health infrastructure and robust AMR bacteria containment efforts. The persistence of high *E.coli* resistance rates [18], despite these measures, may reflect reintroductions via travel or the global food trade [6,7,8,19,20], underscoring the need for coordinated One Health responses.

As international travel grows, so too does the need for targeted risk mitigation. Our data reveal stark geographic differences in travel-related AMR bacteria exposure, reinforcing calls for improved public awareness. Travellers to high-risk regions where resistant bacteria are more readily acquired through food, water or contact should follow CDC-recommended precautions [21]: care with what water is ingested, eat only thoroughly cooked food, avoid raw produce unless properly washed and peeled, practice hand hygiene and use hand sanitiser. These measures not only reduce the risk of illness but also the likelihood of carrying resistant strains home. While most colonisation is transient, some travellers may carry resistant organisms for six months or more [10]. Improved travel advice is needed for both pretravel and post travel. It is especially important for those who are more at risk for adverse consequences from being infected with resistant bacteria, e.g., the elderly, the frail and the immunosuppressed. The data from our model will also be important for improving global surveillance coordination

Travel history remains underutilised in clinical practice, even though it is a critical factor in identifying and managing infections caused by resistant pathogens. This gap is particularly concerning for emergency admissions, post-travel surgeries and inter-hospital transfers, events that have triggered major hospital outbreaks [22,23,24,25,26,27,28].

This analysis has several important limitations and potential confounders that highlight the need for more granular data in future research. We believe that our proxy index used to estimate disease prevalence from mortality data is an innovative approach; however, there are limitations in using death rates as a surrogate for colonization prevalence and its effects. There will be differences between and within advanced countries and poorer resourced countries with mortality rates for common serious infections, e.g., *E.coli* bloodstream infections [29,30]. Deaths are an extreme outcome, but they are common and represent definite and common outcome for serious bacterial infections and can be measured globally much better and accurately than most other potential endpoints.

This is the first paper we are aware of to try and do such an analysis and modelling on a global scale. We chose *E.coli* in our model because it is the most common bacterial pathogen infecting people and causing serious disease globally. *E.coli* has the largest number of isolates available to analyse. We needed a large dataset to be able to do our analysis, as it is a 241 by 241 matrix; this meant we had 58,081 data points and each point needed to have sufficient numbers in them.

In any follow up studies by ourselves, or others, *Staphylococcus aureus* would also be an isolate to examine. But for most other bacteria, although some global data are available, there will not likely be sufficient numbers to do the same type of analysis as we have conducted in this paper.

The Newtonian gravity approach has limitations. If there was a “clean AMR area”, for instance, with less adverse environmental conditions for antibiotic spread, with low population density and with low antibiotic use, this area could be geographically close to an area with high levels of AMR bacterial contamination. We have addressed this issue for individual countries with a population over 5 million within “regions”. Hong Kong is different from many surrounding regions. Even New Zealand is different from Australia, and countries within the EU are markedly different.

Tracking the acquisition and transmission of antimicrobial-resistant (AMR) bacteria at the individual level remains inherently challenging. The scale of global tourism is vast, yet comprehensive data on traveller movements, particularly origin–destination pairs and individual travel itineraries, are sparse or unavailable. While most countries report aggregate arrival and departure figures, detailed bilateral flow data are rare, limiting precision in exposure estimation. Linking disease acquisition to specific destinations is further complicated by multi-country travel, which obscures where colonisation may have occurred. In addition, tourists often engage with environments differently than residents, such as eating in hotels, visiting high-traffic sites or staying in concentrated urban hubs, so national disease prevalence data may not reflect traveller exposure risk accurately. These behavioural distinctions, coupled with variation in duration of stay and contact with local communities, introduce further uncertainty. Our model assumes travel is solely a function of distance and population size. It does not account for crucial confounders like tourism infrastructure, purpose of travel (business vs. leisure), wealth of travellers or seasonality. The exposure risk will also likely differ between tourists and local populations, but we were unable to explore this further. When data becomes available that can incorporate behavioural risk profiles, such as average duration of stay or types of activities (e.g., rural vs. urban travel), this will help to better refine exposure assessments. Visitor touristic behaviour during their stay is important. If the visitor is a resident in a five-star hotel and has limited contact with local people, the possibility of AMR bacterial acquisition would be expected to be lower than in tourism visits that involve not taking any precautions and/or having extensive and prolonged contact with the local people. Other components such as the length of stay in the foreign country (including long stay residencies) or the frequency of visits to high-AMR-infection-containing countries will also have impacts. These, however, are currently not measurable.

The very limited availability of reliable time series data precludes conventional estimation of confidence intervals on this global dataset. The modelling effort is focused on pattern recognition of how travel may spread antimicrobial resistance at a global scale. The discussion and disclaimers point the way to understand the robustness of the analysis, and they highlight the need for future data collection and modelling efforts. We did not include in our model the role of antibiotic consumption in the country of origin of the traveller or in the counties visited. While the resistant organism acquired by travellers may increase in frequency if the home country has an excessive use of antibiotics, many studies have now shown that antibiotic consumption within countries has minimal impact on a global scale to explain differences seen in resistance rates [1,2,3,4,31] What is more important for any impact on a traveller are the levels of antibiotic resistance present in different countries/populations which is impacted by various factors, and this is why we used AMR bacteria rates in our model. We think spread of resistant bacteria is much more important than the volumes of antibiotics consumed in any population re the rate of resistant bacteria seen in that region.

Without higher-resolution data on traveller demographics, routes and exposures, these estimates—while systematically derived—should be interpreted as indicative rather than definitive. We think our conclusions are intriguing but are preliminary and hypothesis-generating rather than definitive because of the many confounders.

Ultimately, while travel offers profound social and economic benefits, it also presents clear challenges for antimicrobial resistance control. Quantifying the risks associated with global mobility is a foundational step toward more effective mitigation—through better-informed travellers, vigilant clinicians and responsive public health systems.

## 4. Methods

### 4.1. Measuring AMR Bacteria Burden Generated by Travel

The risk of a traveller acquiring antimicrobial-resistant (AMR) bacteria while visiting a destination hinges on two primary factors: (a) the prevalence of high-risk bacterial colonization or infection in the host country (e.g., enteric bacteria transmitted via sanitation, food or social contact) and (b) the ambient AMR rate, i.e., the likelihood that any bacterial infection acquired is drug-resistant. A visitor from a country with low colonization prevalence (and thus lower infection rates) poses less risk of transmitting resistant bacteria than a traveller from a country with high colonization levels, even if resistance rates are comparable. The same logic applies to incoming travellers: the risk they pose in introducing AMR bacteria depends on these two factors in their country of origin.

We use *Escherichia coli* resistance to third-generation cephalosporins as our indicator organism for assessing AMR bacteria prevalence, given its status as the most common bacterial cause of serious infections globally. Country-level AMR bacteria data were sourced from the WHO and other contemporary publications [32,33,34,35]. Countries were also aggregated into regions (see Appendix A). For countries without direct data, we applied the average AMR rates of other countries in their respective regions.

Due to limited global surveillance, direct estimates of colonization prevalence are not currently feasible. Instead, we infer disease burden using a proxy index constructed from population-level death rates across four infectious disease categories: enteric infections, tropical diseases, respiratory infections and other infectious diseases [35]. The underlying rationale is that fatal infections are more likely to occur in individuals with high bacterial loads, who are also more prone to shedding and transmission, particularly when symptomatic (e.g., with diarrhoea).

To build this index, we drew on published annual death rate estimates (per 1000 population) for each category, standardizing each series to a mean of 100 and a standard deviation of 15 to reduce noise. The four standardized series were summed, and the combined score was again standardized to the same scale. This index serves as a proxy for colonization-related disease prevalence across countries.

These estimates of antimicrobial resistance rates and inferred infection prevalence by country form the foundation for our comparative assessment of AMR bacteria-related impact of international travel and are illustrated in Figure 1 and Figure 2.

A robust framework for measuring the AMR infection burden linked to international travel must account for at least six critical components:

Outbound travel patterns: Which countries are most frequently visited by residents?

Inbound travel origins: From which countries do foreign visitors arrive?

Travel volume: How many residents of each country travel internationally each year?

Visitor intake: How many foreign visitors does each country receive annually?

Disease prevalence and resistance: What are the national rates of colonizing infectious bacteria and associated antimicrobial resistance?

Population context: How large is each origin and destination population compared to the number of incoming and outgoing travellers?

Exposure impact of travellers (denoted EIT) quantifies the risk via potential exposure that individuals traveling abroad acquire antimicrobial-resistant (AMR) bacteria and subsequently transmit these organisms within their home country upon return. To estimate EIT, we considered each country’s outbound travel volumes, the distribution of destination countries visited and the associated AMR burden of those destinations. The greater the proportion of residents traveling to high-burden regions, the higher the overall exposure risk upon return. Conversely, travel to destinations with low disease prevalence and resistance poses minimal risk. To contextualize this imported exposure, EIT estimates were benchmarked against the home country’s baseline prevalence of infectious disease and antimicrobial resistance. This allows for standardised cross-country comparisons and enables identification of countries where international travel can contribute disproportionately to domestic AMR risk. Let the subindex “*i*” denote the home country of interest and “*j*” be a counter for all other (*N* − 1) countries in our data, where ∑ represents the summation across those (*N* − 1) countries. We define this relationship as follows:(1)$$\begin{array}{cc} & Exposure\;Impact\;of\;Travellers\;returning\;from\;foreign\;countries=EIT_{i}\\ & =\sum_{j\neq i}^{N}\frac{\left(AMR\;rate\;in\;j\right)*\left(Prevalence\;Disease\;in\;j\right)*(Outbound\;Travellers\;from\;i\;to\;j)}{\left(AMR\;rate\;in\;i\right)*\left(Prevalence\;Disease\;in\;i\right)*\left(Population\;i\right)} \end{array}$$

In a similar fashion, exposure impact of visitors (denoted EIV) is defined to measure the risk that visitors, via potential exposure from abroad, transmit antimicrobial-resistant infection to residents of the host country. EIV is defined as follows:(2)$$\begin{array}{cc} & Exposure\;Impact\;of\;Visitors\;from\;foreign\;countries={ERV}_{i}\\ & =\;\sum_{j\neq i}^{N}\frac{\left(AMR\;rate\;in\;j)*(Prevalence\;Disease\;in\;j)*(Inbound\;Visitors\;from\;j\;to\;i\right)}{\left(AMR\;rate\;in\;i\right)*\left(Prevalence\;Disease\;in\;i\right)*\left(Population\;i\right)} \end{array}$$

The overall antibiotic-resistant infection burden of travel is then defined as follows:(3)$$AMR\;Infection\;Burden\;of\;Travel\;for\;country\;i={EIT}_{i}+{EIV}_{i}$$

To deal with the skewness, particularly caused by very large differences in populations and in the amount of travel in and out of countries, all three measures above are reported in logarithmic form as ln(1 + EIT), ln(1 + EIV) and ln(1 + EIT + EIV).

Logarithmic responses to treatments are commonly found in medicine to model the relationships, e.g., between drug concentrations and biological response. For small values of the impacts measured in Equations (1)–(3), the logarithmic values change by nearly by the same amount; however, the logarithmic rate of response becomes more muted as the values rise. For the sake of clarity, Appendix A shows a chart of the relationship between (EIT + EIV) and ln(1 + EIT + EIV) that we subsequently calculate.

### 4.2. Global Travel Patterns

To estimate global travel patterns, we employed a gravity-style trip distribution model, which analogizes Newtonian gravity: the interaction between two locations decreases with distance and increases with the populations of the origin and destination. This approach has been widely used in transportation, trade, migration and epidemiological modelling for over fifty years [36,37,38,39,40,41,42], with additional technical details and references available in the Appendix A.

Resident outbound and visitor inbound travel data were sourced from the World Tourism Statistics Database for 241 countries [43], using 2019 data to avoid distortions caused by the COVID-19 pandemic. Inbound travel data were available for 191 countries and proved more complete than outbound data. Countries missing inbound data were primarily small, low-income island states or territories—unlikely to significantly influence global patterns. To address these gaps, we grouped countries into eighteen geographic regions (Appendix A) and imputed missing values using each region’s average per capita inbound travel rate.

Travel departure counts per capita were reported for 100 countries, accounting for the bulk (87%) of our total estimated global departures. To impute missing outbound travel for the 91 countries with inbound-data only, we fitted an ordinary least-squares regression of outbound departures per capita on GDP per capita (Appendix A) using countries that had outbound data. This equation was then used to estimate outbound data for those 91 countries that had inbound data but not outbound data (Appendix A). For the remaining 50 mostly small countries, we estimated outbound travel based on the average rate of outbound travellers per capita of other countries in their region that had reported outbound data.

Because individuals may visit multiple destinations per trip, total resident departures (1.79 billion) are lower than total inbound visits (2.39 billion). To correct for this, outbound travel figures were scaled up by 33.5% to capture the total number of countries visited.

Travel sensitivity to distance, commonly termed distance elasticity, is defined as the percentage change in a destination’s attractiveness per 1% increase in distance. Nadal and Gallego’s meta-analysis [28], synthesizing 94 studies, yields an average elasticity of 1.01; we adopt a rounded value of 1.0 for this analysis.

Pairwise distances between countries were calculated using the Haversine formula [29], based on capital city coordinates sourced from Dataset Publishing Language [30]. To account for fixed travel initiation costs, distances under 100 km were adjusted to 100 km—affecting only 18 of 28,920 country pairs.

Trip distribution matrices were constructed at the country level. Summary statistics are presented in an 18 × 18 regional flow matrix (Appendix A), while the full 242 × 241 country-pair matrix is available on request.

## Figures and Tables

**Figure 1 antibiotics-14-01055-f001:**
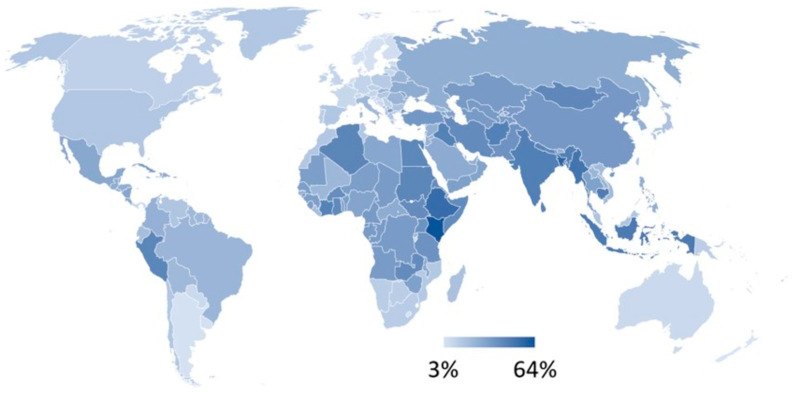
Map of antimicrobial resistance rates of *E.coli* for third generation cephalosporins. Footnote: Country data are available in Appendix A.

**Figure 2 antibiotics-14-01055-f002:**
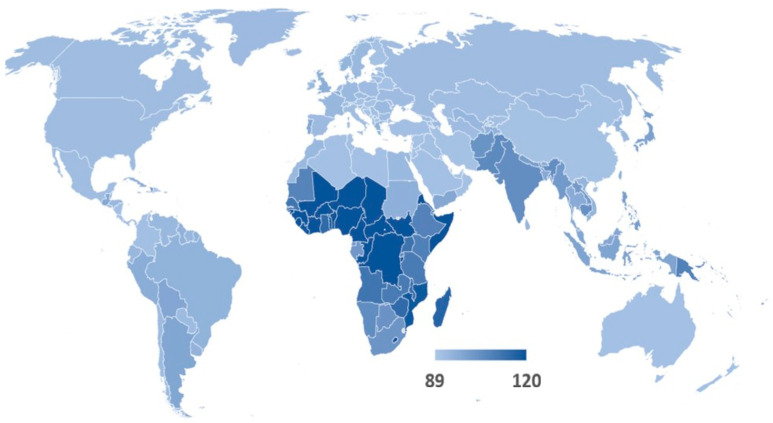
Prevalence of communicable disease proxy index. Footnote: Prevalence proxy index calculation as described in the Section 4 of text. Country data are available in Appendix A.

**Figure 3 antibiotics-14-01055-f003:**
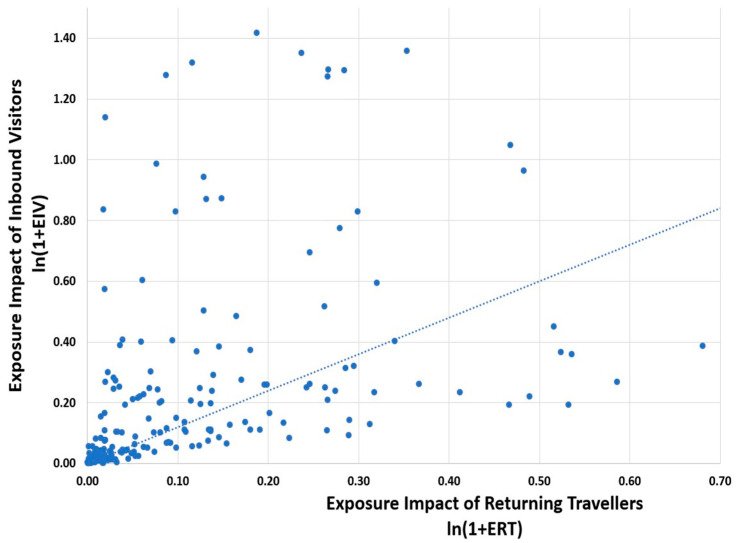
Country-level exposure impact of inbound visitors versus returning travellers. Footnote: Four outliers are not graphed: Andora (0.16, 2.54), Hong Kong (1.8, 1.04), Monaco (0.85, 1.07) and San Marino (0.17 and 2.54). The correlation between the two series is 44%, and the fitted line through the origin of the graph is ln(1 + ERV) = 1.21 × ln(1 + ERT).

**Figure 4 antibiotics-14-01055-f004:**
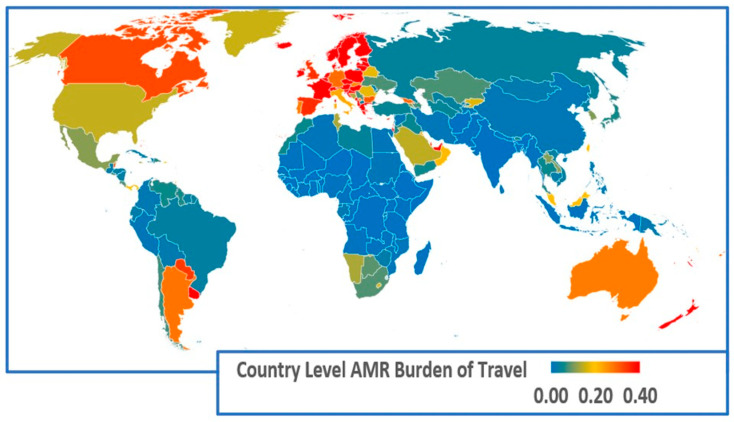
AMR infection burden of travel combined impact of returning resident travellers and arriving visitors {Ln(1 + EIT + EIV)}. Footnote: Estimated AMR infection burden of travel for all countries on a colour scale from 0.01 to 0.4 or greater. AMR infection burden of travel data for all countries are provided in Appendix A.

**Table 1 antibiotics-14-01055-t001:** Regional exposure impact of visitors and returning travellers and the overall AMR infection burden of travel.

Region	Regional Exposure Impact of Returning Travellers	Regional Exposure Impact of Visitors	Regional AMR Infection Burden of Travel
Caribbean	0.05	0.42	0.45
Central America	0.03	0.07	0.11
Central Asia	0.03	0.04	0.07
Eastern Africa	0.01	0.02	0.03
Eastern Asia	0.27	0.32	0.48
Eastern Europe	0.11	0.19	0.27
Middle Africa	0.01	0.00	0.01
Northern Africa	0.01	0.03	0.04
Northern America	0.19	0.26	0.39
Northern Europe	0.36	0.42	0.68
Oceania	0.14	0.34	0.44
South America	0.06	0.04	0.10
South-eastern Asia	0.07	0.17	0.22
Southern Africa	0.08	0.10	0.17
Southern Asia	0.01	0.03	0.04
Southern Europe	0.14	0.58	0.67
Western Africa	0.00	0.01	0.01
Western Asia	0.11	0.11	0.20
Western Europe	0.37	0.35	0.62

The regions with the highest risk of AMR bacteria from returning travellers were Northern Europe and Western Europe. The region with the highest risk of AMR infection from visiting travellers was the Caribbean where small island countries receive large numbers of visitors to their vibrant tourist destinations.

**Table 2 antibiotics-14-01055-t002:** Exposure impact of visitors and returning travellers and overall AMR infection burden of travel: twenty largest impacts of countries with population greater than 10 million people.

Country	Exposure Impact of Returning Travellers	Country	Exposure Impact of Inbound Visitors	Country	AMR Burden of Travel
SWEDEN	0.59	FRANCE	0.50	SWEDEN	0.74
BELGIUM	0.31	CZECH REPUBLIC	0.40	FRANCE	0.58
UNITED KINGDOM	0.27	GREECE	0.37	CZECH REPUBLIC	0.47
NETHERLANDS	0.26	SPAIN	0.30	NETHERLANDS	0.46
GERMANY	0.22	POLAND	0.27	GREECE	0.45
CANADA	0.22	SWEDEN	0.27	BELGIUM	0.41
ARGENTINA	0.19	NETHERLANDS	0.25	POLAND	0.41
AUSTRALIA	0.18	PORTUGAL	0.22	SPAIN	0.35
POLAND	0.17	MALAYSIA	0.15	UNITED KINGDOM	0.35
TAIWAN	0.13	ITALY	0.14	CANADA	0.32
FRANCE	0.13	CANADA	0.13	GERMANY	0.29
ROMANIA	0.12	BELGIUM	0.13	ARGENTINA	0.28
GREECE	0.12	AUSTRALIA	0.11	PORTUGAL	0.28
ITALY	0.11	ARGENTINA	0.11	AUSTRALIA	0.27
SAUDI ARABIA	0.10	UNITED KINGDOM	0.11	ITALY	0.23
CZECH REPUBLIC	0.09	TUNISIA	0.10	MALAYSIA	0.21
UNITED STATES	0.09	GERMANY	0.08	TAIWAN	0.20
SOUTH KOREA	0.07	TAIWAN	0.07	ROMANIA	0.17
SPAIN	0.07	UNITED STATES	0.07	UNITED STATES	0.15
MALAYSIA	0.07	ROMANIA	0.06	SAUDI ARABIA	0.14

To extend the analysis further, Appendix A broadens the scope to include all countries with populations over 5 million, allowing a more comprehensive assessment across 123 countries. In this larger cohort, countries such as Hong Kong Special Administrative Region, Denmark, New Zealand, Hungary, Norway and Sweden emerge at the top of the AMR infection travel risk rankings, with burden of travel scores ranging from 2.06 to 0.74.

## Data Availability

All data used for this study is publicly available via the websites mentioned in our Methods section. Much of the individual country data are summarized in the Appendix A. More data can be supplied on request to the authors in the form of Excel files.

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
