# Peer review of "Tourism and the Global Vectoring of Antimicrobial-Resistant Disease: What Countries Are Most Impacted?"

_antibiotics, 2025, doi:10.3390/antibiotics14111055_

Round 1
Reviewer 1 Report
Comments and Suggestions for Authors
The manuscript presents an important and timely analysis of the role of international travel in the global dissemination of antimicrobial resistance (AMR). The following points are suggested to improve the clarity, rigor, and impact of the study:
While the use of a gravity-style trip distribution model is well-established, additional details on the calibration and validation of this model would improve confidence in the travel estimates. For instance, specifying how missing outbound data were imputed for countries lacking reports would enhance reproducibility.
The proxy index used to estimate disease prevalence from mortality data is an innovative approach; however, the limitations of using death rates as a surrogate for colonization prevalence should be discussed more explicitly. Including sensitivity analyses or alternative proxies could strengthen the robustness of the results.
The selection of E. coli resistance to third-generation cephalosporins is justified, yet the manuscript could benefit from a brief rationale on why other important organisms (e.g., Klebsiella pneumoniae, Acinetobacter baumannii) were not included.
The exposure impact metrics (EIT and EIV) are well-defined mathematically, but the public health significance of their logarithmic transformations is not always intuitive. Clarifying the real-world meaning of a one-unit change in these values would improve accessibility for a broader readership.
While regional-level aggregation provides stability, it may obscure important intra-regional heterogeneity. Highlighting selected country-level outliers or contrasting high-burden and low-burden countries within the same region could yield deeper insights.
The manuscript acknowledges that exposure risk may differ between tourists and local populations, but does not explore this further. Incorporating behavioral risk profiles, such as average duration of stay or types of activities (e.g., rural vs. urban travel), could refine exposure assessments.
The discussion briefly touches on the need for a One Health approach and improved travel advisories, but the policy implications of the findings could be expanded. For example, how might these data inform pre-travel counseling, airport screening, or global surveillance coordination?
The limitations section is appropriately cautious, but the potential impact of imputation methods, underreporting, or inconsistencies across surveillance systems deserves further elaboration. Uncertainty bounds or confidence intervals on key findings would help quantify this.
The maps and regional tables are effective. Nonetheless, including interactive or downloadable country-level data summaries (as supplementary material) would enhance usability for researchers and policymakers.
Several typographical and formatting inconsistencies are present (e.g., “region with where” in line 48). A thorough proofreading is recommended before publication. In addition, please write bacterial names correctly.
Author Response
Please see below attached.

Reviewer 2 Report
Comments and Suggestions for Authors
This is highly informative and interesting manuscript focusing the risks of acquisition of antibiotic resistant bacteria by traveling activities.
Line 114. At the very least, the authors should mention the possibility of including other components in the future. For instance "Visitor touristic behavior during stay". If in the travel the visitor is resident in a five stars hotel, and has no intimate relations with local people, the possibility of AMR acquisition should be shorter than in tourism visiting the "deep country" without major precautions, and/or having promiscuity with local people. Other component are the length of stay in the foreign country (including long stay residencies), or the frequency of visits to high AMR containing countries.
Line 135. The authors should at least mention the opposite possibility, that is , the EIT to susceptible bacteria when visiting AMR "clean" countries. Hypothetically susceptible bacteria should have, in the absence of antibiotic exposure, a higher fitness and could replace resistant microorganisms. If these components cannot be extracted from Tourism databases, it should be adverted, as it probably influences the spread of AMR.
Line 143. I am not against the Newtonian gravity approach, but the limitations should be considered, as a "clean AMR area", for instance, with adverse environmental conditions for antibiotic spread, low population density and low antibiotic use, could be geographically close to an AMR contaminated area. Perhaps biogeographical regions should be included in a future model.
Line 189. The utility of the Index of Communicable Disease is not totally convincing, as most of these communicable diseases are not targets for therapy with third-generation cephalosporins.
Final comment: Nothing is mentioned about the role of antibiotic consumption in the country of origin of the voyager. The resistant organism acquired by travelers may increase in frequency if the home country has an excessive use of antibiotics. I know that “Antibiotics” do not like that reviewers suggest their own references to the authors, but in this case, recently a paper on the relative role of “selection and transmission” just focusing on the incidence of third-generation cephalosporins in various countries. In Scandinavian countries, the role of the rate of third-generation consumption practically has no impact on the increase of resistance, as the sanitation is very good (Baquero, F., Pérez-Cobas, A. E., Aracil-Gisbert, S., Coque, T. M., Zamora, J. 2024). Selection versus transmission: quantitative and organismic biology in antibiotic resistance. Infection, Genetics and Evolution, 121, 105606). At your will.
Perhaps it is worth mentioning that the differential risks detected in this manuscript should be particularly taken into account by elderly, frail travelers, where the acquisition of antibiotic resistance may constitute a major obstacle for therapy.
Author Response
- This is highly informative and interesting manuscript focusing the risks of acquisition of antibiotic resistant bacteria by traveling activities.
We thank the reviewer for the very supportive comments made on the importance of our paper and for the many helpful suggestions the reviewer made to improve it further.
- Line 114. At the very least, the authors should mention the possibility of including other components in the future. For instance, "Visitor touristic behavior during stay". If in the travel the visitor is resident in a five stars hotel, and has no intimate relations with local people, the possibility of AMR acquisition should be shorter than in tourism visiting the "deep country" without major precautions, and/or having promiscuity with local people. Other component are the length of stay in the foreign country (including long stay residencies), or the frequency of visits to high AMR containing countries.
We agree this is a cofounder and added these points to the discussion at line 348 to 370. However, the extra data suggested is currently not available to examine this on a global scale, so we cannot add that type of analysis currently. In future research we think that is worth pursuing.
Line 135. The authors should at least mention the opposite possibility, that is, the EIT to susceptible bacteria when visiting AMR "clean" countries. Hypothetically susceptible bacteria should have, in the absence of antibiotic exposure, a higher fitness and could replace resistant microorganisms. If these components cannot be extracted from Tourism databases, it should be adverted, as it probably influences the spread of AMR.
We agree and have added a comment in the discussion at line 294 to 297
- Line 143. I am not against the Newtonian gravity approach, but the limitations should be considered, as a "clean AMR area", for instance, with adverse environmental conditions for antibiotic spread, low population density and low antibiotic use, could be geographically close to an AMR contaminated area. Perhaps biogeographical regions should be included in a future model.
We agree and have added a comment in the discussion at line 340 to 346. It highlights the reason why we also looked at individual countries with population of over 5 million within “regions”. Hong Kong is different to many surrounding regions, Even New Zealand is different to Australia and countries within the EU markedly different.
- Line 189. The utility of the Index of Communicable Disease is not totally convincing, as most of these communicable diseases are not targets for therapy with third-generation cephalosporins.
We believe while it has limitations, it the best index we could put together based on available global data. We have added a point about its limitation in the discussion on line 322 to 330.
- Final comment: Nothing is mentioned about the role of antibiotic consumption in the country of origin of the voyager. The resistant organism acquired by travellers may increase in frequency if the home country has an excessive use of antibiotics. I know that “Antibiotics” do not like that reviewers suggest their own references to the authors, but in this case, recently a paper on the relative role of “selection and transmission” just focusing on the incidence of third-generation cephalosporins in various countries. In Scandinavian countries, the role of the rate of third-generation consumption practically has no impact on the increase of resistance, as the sanitation is very good (Baquero, F., Pérez-Cobas, A. E., Aracil-Gisbert, S., Coque, T. M., Zamora, J. 2024). Selection versus transmission: quantitative and organismic biology in antibiotic resistance. Infection, Genetics and Evolution, 121, 105606). At your will.
We agree that antibiotic consumption is likely important but on a global scale has much less impact than commonly assumed when measured against resistance rates, as we and others have shown in previous research (refs 1-4). We think spread of resistant bacteria is much more important than the volumes of antibiotics consumed in any population re the rate of resistant bacteria seen in that region. We have published numerous papers trying to point this - often with reviewers not agreeing with us!.
We thank the reviewer for pointing out this additional paper that also show this lack of associate with resistance and antibiotic consumption, and we have added them to our reference list and added a comment on the importance of antibiotic consumption at line 376 to 3875
The rates of resistance seen in any country will be the result of many factors that includes antibiotic consumption. We in this paper have looked at the important end point i.e. resistance rates rather than the contributors to what causes resistance to assess the risk for travellers. We therefore don’t think we need to separately look at antibiotic consumption rates in each country.
- Perhaps it is worth mentioning that the differential risks detected in this manuscript should be particularly taken into account by elderly, frail travellers, where the acquisition of antibiotic resistance may constitute a major obstacle for therapy.
We agreed and have added comments in the discussion at line 313 to 316 as it involves pre-travel counselling as suggested also by reviewer 1.
Reviewer 3 Report
Comments and Suggestions for Authors
Tourism and the global vectoring of antimicrobial resistant disease: what countries are most impacted?
This study by Collignon and Beggs attempted to model the global vectoring of Antimicrobial Resistance (AMR) through international travel. The core concept is strong, and the scale of the analysis (241 countries) is ambitious. However, the methodological approach, while innovative, contains significant assumptions and simplifications that substantially weaken the robustness and interpretability of the findings. The conclusions are intriguing but should be presented as preliminary and hypothesis-generating rather than definitive as presented in the current manuscript.
Major Comments:
1. The methodological approach, while innovative, contains significant assumptions and simplifications that substantially weaken the robustness and interpretability of the findings.
2. While the objective to create a “regional and country level indices measuring the impact AMR” is clear. However, the term “impact” is misleading throughout the paper. The study models relative exposure potential, not a measurable health or economic “impact” (e.g., increased incidence of resistant infections, mortality, cost to healthcare systems). This conflation is a major weakness.
3. Using death rates from infectious diseases as a proxy for the prevalence of colonizing bacteria is a highly problematic and unsupported assumption. Death is an extreme outcome. A country can have a high prevalence of colonizing bacteria (e.g., resistant E. coli in the gut) with modern healthcare leading to low death rates. Conversely, death rates can be high due to factors unrelated to colonization prevalence (e.g., HIV prevalence affecting TB outcomes, lack of access to basic care). This proxy likely misclassifies many countries and is the model's greatest weakness.
4. The model assumes travel is solely a function of distance and population size. It does not account for crucial confounders like tourism infrastructure, purpose of travel (business vs. leisure, VFR), wealth of travellers, or seasonality. Using a single distance elasticity value for all country pairs is a vast oversimplification.
5. Imputing missing AMR and travel data using regional averages introduces significant noise and potential bias, especially for countries that are outliers within their regions.
6. The main issue is the interpretation of the results. The indices EIT and EIV are unitless, relative metrics. Stating that Hong Kong has a burden of "2.06" and Sweden has "0.74" is not meaningful on an absolute scale. The results show which countries have a high potential exposure ratio based on the model's assumptions, not which countries are "most impacted" in a real-world sense. The ranking is sensitive to the flawed prevalence proxy.
7. The discussion of limitations (Lines 275-290) is inadequate. It focuses on data granularity (which is acknowledged) but fails to critically address the fundamental methodological limitations, specifically: (i) the validity of using death rates as a proxy for colonization prevalence, (ii) The oversimplifications inherent in the gravity model and (iii) the conflation of “exposure potential” with “burden” or “impact.” These unacknowledged limitations severely qualify the conclusions.
8. The conclusions are not fully supported by the data. The model suggests interesting patterns, but the underlying assumptions are too weak to state with confidence that the identified countries are definitively the “most impacted.”
Minor Comments:
- The abstract overstates the findings by claiming to measure “impact” and “risk,” terms that imply a quantified effect on health outcomes, which the model does not actually provide. It models a potential exposure risk index, not actual health impact.
- Line 25–27: Claim that Hong Kong, Denmark, etc., are the “highest risk” is not contextualized with confidence intervals or probability ranges. Without uncertainty measures, this statement risks overstating certainty.
- Line 88–89: Using regional averages to impute missing data introduces strong ecological fallacy risk, differences between countries in the same region can be orders of magnitude.
- Originality is partial. Prior studies (cited refs. 9–16) already documented travel-related AMR.
- Line 95-97: "The underlying rationale is that fatal infections are more likely to occur in individuals with high bacterial loads, who are also more prone to shedding and transmission, particularly when symptomatic (e.g., with diarrhoea)." This is a logical leap. While this may be true at an individual level, it does not logically follow that national death rates are a proportional proxy for national colonization prevalence or shedding rates. This is the core unsupported assumption.
- Line 128-132: The definitions for EIT and EIV are clear. However, the denominator (AMR rate in t) * (Prevalence Disease in t) * (Population t) is confusing. It seems intended to normalize the index to the domestic situation, but it creates a counterintuitive outcome: a country with a high domestic AMR rate and disease prevalence will appear to have a lower "impact" from travel, all else being equal. This may not reflect reality, as a high domestic burden could be exacerbated by imported novel resistances.
- Line 259: “The resistant bacteria they then carry are most likely acquired from the ‘environment’.” This is too categorical. Probabilistically, “likely” should be qualified with data (percent estimates or odds ratios).
- Over-reliance on the authors’ own prior work (Collignon & Beggs cited multiple times), which risks confirmation bias. More independent validation studies should be cited.
Based on the major methodological flaw, I cannot recommend this manuscript for publication.
Author Response
Please see below attached.

Reviewer 4 Report
Comments and Suggestions for Authors
This is a really interesting paper that will be of relevance to anyone working in the field of AMR. I have only a few minor comments.
Abstract. I found the abstract easier to follow AFTER I had read through the paper, which rather defeats the purpose of the abstract. Could you please make it clearer in the abstract that you estimate 1) the impact of returning travellers and 2) visiting travellers and COMBINE these to give an overall AMR travel impact. This wasn’t clear to me until after I read the paper.
L78. You state “(b) the ambient AMR rate, i.e., the likelihood that any bacterial infection acquired is drug-resistant.” Do you mean “infection” or “infection OR colonisation” because in the preceding point a) you refer to colonisation.
Fig 1 and Fig 2. While the shades of blue allow for incremental variations of rates to be demonstrated across countries and regions, I found it visually difficult to interpret. I wonder if categories of low, moderate, high and very high – with different colours (as used in Fig 4) might be easier.
Author Response
- This is a really interesting paper that will be of relevance to anyone working in the field of AMR. I have only a few minor comments.
Thank you for your supportive comments
- I found the abstract easier to follow AFTER I had read through the paper, which rather defeats the purpose of the abstract. Could you please make it clearer in the abstract that you estimate 1) the impact of returning travellers and 2) visiting travellers and COMBINE these to give an overall AMR travel impact. This wasn’t clear to me until after I read the paper.
Thank you. We have altered the abstract to try and make your point clear.
- You state “(b) the ambient AMR rate, i.e., the likelihood that any bacterial infection acquired is drug-resistant.” Do you mean “infection” or “infection OR colonisation” because in the preceding point a) you refer to colonisation.
We mean infection, but colonisation will precede the infection. We have altered the text in the preceding point to add infection to remove this potential confusion. See line 78.
Fig 1 and Fig 2. While the shades of blue allow for incremental variations of rates to be demonstrated across countries and regions, I found it visually difficult to interpret. I wonder if categories of low, moderate, high and very high – with different colours (as used in Fig 4) might be easier.
Thank you. For those wanting more fine-grained information, the country level data used for these figures are reported for all 241 countries in Supplementary Table S2. We prefer to leave Figures 1 and 2 on a grey-scale format so as not to take focus away from the key results in Table 1 and the coloured map in Figure 4.
Round 2
Reviewer 1 Report
Comments and Suggestions for Authors
The authors have thoroughly addressed the comments and suggestions provided during the previous round of peer review. They have clarified methodological details, such as the imputation of missing travel data and the construction of the disease prevalence proxy index. Additionally, the rationale for selecting E. coli as the indicator organism is well-justified and appropriately explained in the revised manuscript. The limitations related to data availability, behavioral heterogeneity among travelers, and the use of logarithmic transformations have been acknowledged and discussed with greater clarity.
Moreover, the policy implications have been strengthened, particularly regarding pre-travel counselling and the potential use of findings for global AMR surveillance strategies. The revisions significantly improve the clarity and robustness of the manuscript.